

# Comparison of visual quality after wavefront-guided LASIK in patients with different levels of preoperative total ocular higher-order aberrations: a retrospective study

Yu Zhang[1,2,*], Yangrui Du[1,2,*], Ming He[1], Youdan Zhang[1] and Zhiyu Du[1]

[1] Department of Ophthalmology, The Second Affiliated Hospital of Chongqing Medical University, Chongqing, China
[2] Medal Eye Institute, Chongqing, China
* These authors contributed equally to this work.

Corresponding author
Zhiyu Du,
duzhiyu@hospital.cqmu.edu.cn

## ABSTRACT

**Purpose:** To compare the visual quality after wavefront-guided femtosecond LASIK (WFG FS-LASIK) in patients with different levels of preoperative total ocular higher-order aberrations to guide clinical decision-making regarding patient selection and treatment strategies.

**Methods:** This study included 112 right eyes of 112 patients who previously underwent WFG FS-LASIK for correcting myopia and myopic astigmatism. The patients were divided into two groups based on the mean values of preoperative total ocular HOAs ($0.30 \pm 0.09$ μm): HOA ≤ 0.3 and > 0.3 groups. The visual acuity, manifest refraction, corneal Strehl ratio (SR), root mean square (RMS) of corneal and ocular aberrations, and area under the log contrast sensitivity function (AULCSF) of both groups were compared preoperatively and at 1, 3, 6, and 12 months postoperatively.

**Results:** The induced ocular HOAs and coma ($\Delta$ = 1 mo − Preop) were significantly lower in the HOAs > 0.3 group than in the HOAs ≤ 0.3 group ($\Delta$HOAs: $0.39 \pm 0.19$ vs. $0.29 \pm 0.18$ μm, $t = 2.797$, $P = 0.006$; $\Delta$ coma: $0.30 \pm 0.19$ vs. $0.20 \pm 0.21$ μm, $t = 2.542$, $P = 0.012$). In the HOAs > 0.3 group, $\Delta$HOAs were negatively correlated with the preoperative ocular HOAs ($r = -0.315$, $P = 0.019$). In the HOAs ≤ 0.3 group, the regression equation for $\Delta$ HOAs = 0.098 + 0.053 |SE| ($F = 21.756$, $P < 0.001$). In the HOAs > 0.3 group, the regression equation for $\Delta$HOAs = 0.534 − 1.081 HOAs + 0.038|Sphere| ($F = 7.954$, $P = 0.001$). The postoperative uncorrected distance visual acuity, spherical equivalent, corneal aberrations, SR and AULCSF of both groups were similar (all $P > 0.05$). Furthermore, the ocular aberrations were not significantly different between both groups at 3, 6, and 12 months postoperatively (all $P > 0.05$). In addition, compared with the preoperative period, the AULCSF of both groups were significantly increased in the postoperative period (all $P < 0.05$).

**Conclusions:** The induced ocular HOAs and coma in HOAs > 0.3 group were lower. However, both groups achieved equivalent and excellent visual quality after WFG FS-LASIK. WFG FS-LASIK may provide significant visual benefits for a wider range of patients.

## INTRODUCTION

Owing to its safety, predictability, and effectiveness, laser *in situ* keratomileusis (LASIK) is the most commonly performed refractive surgery (*Wen et al., 2017*; *Taneri et al., 2022*; *Kamiya et al., 2017*; *Chua et al., 2019*). Previous studies have reported that an increase in higher-order aberrations (HOAs) associated with conventional LASIK can lead to glare, halos, and starbursts in night vision. Furthermore, studies have reported that compared with conventional LASIK treatment, wavefront-guided femtosecond LASIK (WFG FS-LASIK) can induce fewer adverse visual symptoms and result in better vision performance (*Gui et al., 2021*; *Zheng et al., 2016*). This may be attributed to its customized ablation pattern for minimizing pre-existing HOAs, while the robust eye tracking and iris-registration system limits the surgically induced HOAs (*Wu et al., 2013*; *Khalifa, El-Kateb & Shaheen, 2009*).

Several studies have suggested that wavefront-guided technology is the most beneficial for patients with higher preoperative HOAs (*Shao et al., 2022*; *Jahadi Hosseini, Abtahi & Khalili, 2016*; *Valentina et al., 2015*). In other words, the benefits of wavefront-guided technology are not very obvious in patients with myopia and lower preoperative HOAs. Because the changes in total HOAs had a statistically significant negative correlation with its preoperative value, meaning that the lower the preoperative HOAs, the higher increase in the postoperative value (*Jahadi Hosseini, Abtahi & Khalili, 2016*). Even some studies have suggested that the wavefront-guided technology is not needed for patients with preoperative ocular HOAs of <0.30 μm (*Stonecipher & Kezirian, 2008*; *Stonecipher, Parrish & Stonecipher, 2018*). However, these speculations lack of relevant studies on visual quality. Therefore, what about visual quality after WFG FS-LASIK for patients with preoperative ocular HOAs of <0.30 μm? Will different levels of preoperative ocular HOAs lead to different visual qualities after WFG FS-LASIK? Based on clinical observations, we hypothesize that preoperative ocular HOAs may minimally impact visual quality after WFG FS-LASIK. Nevertheless, no clinical studies on this issue are available. In this retrospective study, we compared the visual quality after WFG FS-LASIK in patients with different levels of preoperative total ocular higher-order aberrations to guide clinical decision-making regarding patient selection and treatment strategies. If all patients could achieve equivalent and excellent visual quality outcomes, it suggests that WFG FS-LASIK may provide significant visual benefits for a wider range of patients, even those with relatively low preoperative HOAs.

In this study, visual acuity, manifest refraction, corneal Strehl ratio, the root mean square (RMS) values of corneal and ocular aberrations, and contrast sensitivity function (CSF) were used to assess visual quality. Although the subjective visual scale is also an important research indicator of visual quality, since it reflects the visual quality of both eyes of patients, and many patients exhibit varying levels of higher-order aberrations in each of

their eyes, which may interfere with the results of the subjective visual scale. Therefore, the subjective visual scale was not included in our study.

## PATIENTS AND METHODS

### Patients

In this retrospective, non-randomized cohort study, we evaluated 112 right eyes from 112 patients who underwent WFG FS-LASIK at Chongqing Medal Eye Institute between June 2018 and October 2020. All patients' preoperative, 1, 3, 6, and 12 months postoperative data were complete without missing.

The inclusion criteria were as follows: participants aged 18 to 40 years, corrected distance visual acuity (CDVA) of 20/20 or better, and stable refraction for >1 year. Patients who were wearing rigid contact lenses were instructed to stop wearing them at least 4 weeks preoperatively, whereas those who were wearing soft contact lenses were instructed to stop wearing them within the previous 2 weeks. The exclusion criteria were as follows: patients with diabetes mellitus or autoimmune diseases; those with a history of ocular surgery, trauma, or ocular diseases other than myopia or astigmatism; and those who were nursing or pregnant.

As several studies have suggested that the wavefront-guided technology may be more appropriate for patients who have preoperative RMS of ocular HOAs >0.3 μm (*Valentina et al., 2015*; *Feng, Yu & Wang, 2011*). Clinically, 0.3 μm is also commonly used as the cut-off point for whether a patient is suitable for wavefront aberration surgery. The mean value of the preoperative RMS of ocular HOAs in our study was 0.30 ± 0.09 μm (range 0.10–0.63 μm). Therefore, we divided the patients into two groups based on the mean value of preoperative RMS of ocular HOAs: HOAs ≤ 0.3 μm and HOAs > 0.3 μm. This grouping allowed us to compare the visual outcomes of WFG FS-LASIK in patients with different levels of preoperative aberrations.

This study was approved by the Ethics Committee of the Second Affiliated Hospital of Chongqing Medical University (No. 76/2022). Patients were informed about study inclusion and provided written informed consent.

## METHODS

Ophthalmologic examinations, including logMAR of uncorrected distance visual acuity (UDVA), logMAR of CDVA, slit-lamp microscopic examination of the anterior segments, indirect retinoscopy of the posterior segments, refraction measurements with and without cycloplegia, intraocular pressure (IOP) measurements using a noncontact tonometer (AT555; Rerchert Inc., Depew, NY, USA), corneal curvature (ACCUREF-K9001; Shin-Nippon Inc., Tokyo, Japan), corneal thickness using an ultrasound pachymeter (300AP+; Sonomed Inc., Lake Success, NY, USA), corneal topography (Sirius, CSO Inc., Cosenza, Italy), wavefront aberrations (Wavescan Vision 3.68; VISX Inc., Santa Clara, CA, USA), and contrast sensitivity (CS) (CSV-1000E; VectorVision Inc., Greenville, OH, USA), were performed preoperatively and at 1, 3, 6, and 12 months postoperatively.

Wavefront aberration serves as a sensitive and comprehensive evaluation index for assessing the overall and component optical performance of the human eye. In general, an increase in wavefront aberration is closely associated with a decline in visual quality. As WFG FS-LASIK is performed on the cornea, the changes in corneal aberrations need to be particularly noted. Corneal topography (Sirius, CSO Inc., Cosenza, Italy) was performed to identify corneal aberrations in a 6-mm zone. A skilled technician performed at least triplicate measurements on the same eye. The following parameters were analyzed and recorded: Strehl ratio (SR), RMS of total corneal aberrations (TCAs), astigmatism, coma, trefoil, spherical aberrations, and HOAs.

The ocular wavefront aberrations are the combined outcome of corneal and intraocular aberrations, offering a comprehensive assessment of the ocular optical performance. Based on the principle of the Hartmann–Shack wavefront sensor technique, the WaveScan Wavefront aberrometer (Wavescan Vision 3.68; VISX Inc., Santa Clara, CA, USA) was used to measure ocular wavefront aberrations. A skilled technician performed at least triplicate measurements on the same eye with a 6.0 mm pupil diameter. The following parameters were analyzed and recorded: RMS of total ocular aberrations (TOAs), defocus, astigmatism, coma, trefoil, spherical aberrations, and HOAs.

The contrast sensitivity assesses the eye's capacity to discern visual targets at varying contrasts, offering a more comprehensive evaluation of visual function compared to a standard vision test. Monocular best-corrected distance CS was evaluated under scotopic (3 cd/m$^2$) condition, photopic (85 cd/m$^2$) with and without glare conditions at four spatial frequencies (3, 6, 12, and 18 cycles per degree, c/d). The CS log values at each spatial frequency were recorded, and the area under the log contrast sensitivity function (AULCSF) was calculated for data analysis. The higher the AULCSF value, the greater the visual quality.

## Surgical procedure

An experienced surgeon performed all surgical procedures. For the LASIK procedure, a superior-hinged corneal flap was created using a 60-kHz Intralase iFS femtosecond laser (AMO Inc., Santa Ana, CA, USA) with a flap diameter of 8.5 mm and a thickness of 100 μm. The WaveScan System aberrometer and Visx CustomVue Star S4 IR excimer laser (AMO Inc., Santa Ana, CA, USA) with a planned optical zone diameter of 6.0 mm and an ablation zone of 8.5 mm were used to perform WFG treatments. First, the flap was repositioned; then, the interface was irrigated with a balanced saline solution. After refractive surgery, tobramycin 0.3% ophthalmic solution (Tobrex, Alcon Inc., Bornem, Belgium) and fluorometholone 0.1% ophthalmic solution (Fluorometholone, Allergan Inc., Dublin, Ireland) were administered four times daily for 7 days and the first 2 weeks, respectively. The dose of the fluorometholone 0.1% ophthalmic solution was gradually tapered, decreasing the frequency of administration to every 2 weeks (three times daily, two times daily, and finally once daily). Finally, 0.1% sodium hyaluronate eye drops (HYCOSAN, URSAPHARM Inc., Saarbrücken, Germany) were administered four times daily for 1 month.

## Statistical analysis

SPSS software (version 26.0; IBM Inc., Armonk, NY, USA) was used to perform statistical analysis. Data normality was confirmed using the Kolmogorov–Smirnov test. Normally distributed data were represented as mean ± standard deviation. The independent samples t-test and repeated measures analysis of variance were used to analyze the data consistent with normal distribution and homogeneity of variance. Pearson's correlation and stepwise multiple linear regression analyses were performed to analyze the possible factors affecting ocular HOA induction. $\chi^2$ analysis was performed to compare the proportions. In case the data were not normally distributed, the Wilcoxon Mann–Whitney test was performed. A P-value of <0.05 was considered statistically significant.

## RESULTS

In this study, 112 right eyes of 112 patients were included. The baseline data of both groups are presented in Table 1. No significant differences were observed in these variables between the two groups (all $P > 0.05$).

### Efficacy and safety

At 12 months after WFG FS-LASIK, both groups exhibited excellent UDVA, with no patient having a postoperative CDVA of <20/20. No significant differences in the mean UDVA and number of eyes achieving a specified UDVA, for example, 20/16 or better, were observed between the two groups (all $P > 0.05$) (Fig. 1A). Furthermore, no eye lost one or more lines in CDVA (Fig. 1B). The postoperative UDVA gradually improved in both groups. For the HOAs ≤ 0.3 group, the 6-month postoperative UDVA was better than the preoperative CDVA ($-0.08 \pm 0.05$ vs. $-0.10 \pm 0.06$, $Z = -2.117$, $P = 0.034$). Furthermore, for the HOAs > 0.3 group, the 3-month postoperative UDVA was better than the preoperative CDVA ($-0.08 \pm 0.05$ vs. $-0.10 \pm 0.06$, $Z = -2.449$, $P = 0.014$).

At 12 months postoperatively, the mean efficacy index (postoperative UDVA–preoperative CDVA ratio) was $1.01 \pm 0.01$ and $1.00 \pm 0.01$ in the HOA ≤ 0.3 and HOA > 0.3 groups, respectively ($Z = -1.142$, $P = 0.254$). The mean safety index (postoperative CDVA–preoperative CDVA ratio) was $1.01 \pm 0.01$ and $1.01 \pm 0.01$ in the HOA ≤ 0.3 and HOA > 0.3 groups, respectively ($Z = -0.720$, $P = 0.471$).

### Refractive error, predictability, and stability

No significant differences were observed between the attempted spherical equivalent (SE) and the achieved SE between both groups ($0.10 \pm 0.55$ and $0.02 \pm 0.53$ D for the HOAs ≤ 0.3 and HOAs > 0.3 groups, respectively, $t = 0.837$, $P = 0.404$) (Figs. 1C and 1D). At 12 months postoperatively, the SE of 94.74% (54/57) of the eyes in the HOAs ≤ 0.3 group and 87.27% (48/55) of those in the HOA > 0.3 group was within ±1.00 D; however, no significant difference was observed between both groups ($\chi^2 = 1.918$, $P = 0.166$, Fig. 1E). Furthermore, in both groups, postoperative astigmatism was within 0.50 D (Fig. 1F).

The postoperative SE of both groups exhibited a gradual decreasing trend (Fig. 1G). From 3 to 12 months postoperatively, the SE of the HOAs ≤ 0.3 group decreased by $0.19 \pm 0.28$ D ($t = 5.016$, $P < 0.001$), whereas that of the HOAs > 0.3 group decreased by $0.10 \pm$

**Table 1 Preoperative characteristics (mean ± SD, range).**

| Variable | HOAs ≤ 0.3 group | HOAs > 0.3 group | *P* value |
|---|---|---|---|
| Number of eyes | 57 | 55 | – |
| Age (years) | 24.65 ± 5.53 (18 to 40) | 24.91 ± 6.64 (18 to 40) | 0.822 |
| Gender (% female) | 71.90% | 56.40% | 0.087 |
| Sphere (D) | −4.62 ± 1.88 (−0.25 to −9.25) | −4.00 ± 1.78 (−1.25 to −7.50) | 0.077 |
| Cylinder (D) | −0.95 ± 0.72 (0.00 to −2.75) | −1.05 ± 0.78 (0.00 to −3.25) | 0.512 |
| Spherical equivalent (D) | −5.09 ± 1.99 (−0.88 to −10.13) | −4.52 ± 1.88 (−1.63 to −8.38) | 0.121 |
| UDVA (LogMAR) | 1.15 ± 0.31 (0.40 to 1.70) | 1.08 ± 0.34 (0.30 to 1.70) | 0.301 |
| CDVA (LogMAR) | −0.08 ± 0.05 (−0.20 to 0.00) | −0.08 ± 0.05 (−0.20 to 0.10) | 0.612 |
| CCT (μm) | 541.25 ± 26.71 (486 to 605) | 550.15 ± 25.65 (503 to 603) | 0.075 |
| NCT (mmHg) | 13.06 ± 2.19 (10.00 to 18.50) | 13.30 ± 2.36 (10.00 to 19.00) | 0.569 |
| Pupil diameter (mm) | 6.75 ± 0.71 (5.04 to 8.05) | 6.75 ± 0.62 (5.39 to 8.17) | 0.977 |
| Optical zone (mm) | 6.40 ± 0.25 (5.80 to 6.80) | 6.48 ± 0.28 (5.70 to 6.80) | 0.125 |

**Note:**
SD, standard deviation; UDVA, uncorrected distance visual acuity; logMAR, logarithm of the minimum angle resolution; CDVA, corrected distance visual acuity; CCT, central corneal thickness; NCT, non-contact tonometer; D, diopter.
$P < 0.05$ statistically significant.

0.26 D ($t = 3.032$, $P = 0.004$). Nevertheless, no significant difference was observed between the two groups ($t = 1.318$, $P = 0.190$). The proportion of the eyes with SE changes of >0.50 D was 3.51% (2/57) in the HOAs ≤ 0.3 group and 5.45% (3/55) in the HOAs > 0.3 group, with no statistical significance between both groups ($\chi^2 = 0.248$, $P = 0.618$).

## Corneal aberrations

As demonstrated in Table 2, before WFG FS-LASIK, the RMS of corneal HOAs and trefoil were higher in the HOAs > 0.3 group than in the HOAs ≤ 0.3 group ($t = −2.87$, $P = 0.005$; $t = −2.22$, $P = 0.029$, respectively). However, no significant differences were observed in the preoperative values of SR, TCAs, astigmatism, coma, trefoil, and spherical aberrations between both groups (all $P > 0.05$). At 1, 3, 6, and 12 months postoperatively, both groups had similar corneal aberrations and change values (Δ = 1 mo − Preop) (all $P > 0.05$).

## Ocular aberrations

The preoperative RMS of ocular HOAs, coma, trefoil, and spherical aberrations were significantly higher in the HOAs > 0.3 group than in the HOAs ≤ 0.3 group (all $P < 0.001$, Table 3). Except for the astigmatism and trefoil at 1 month postoperatively, no significant

Peerj

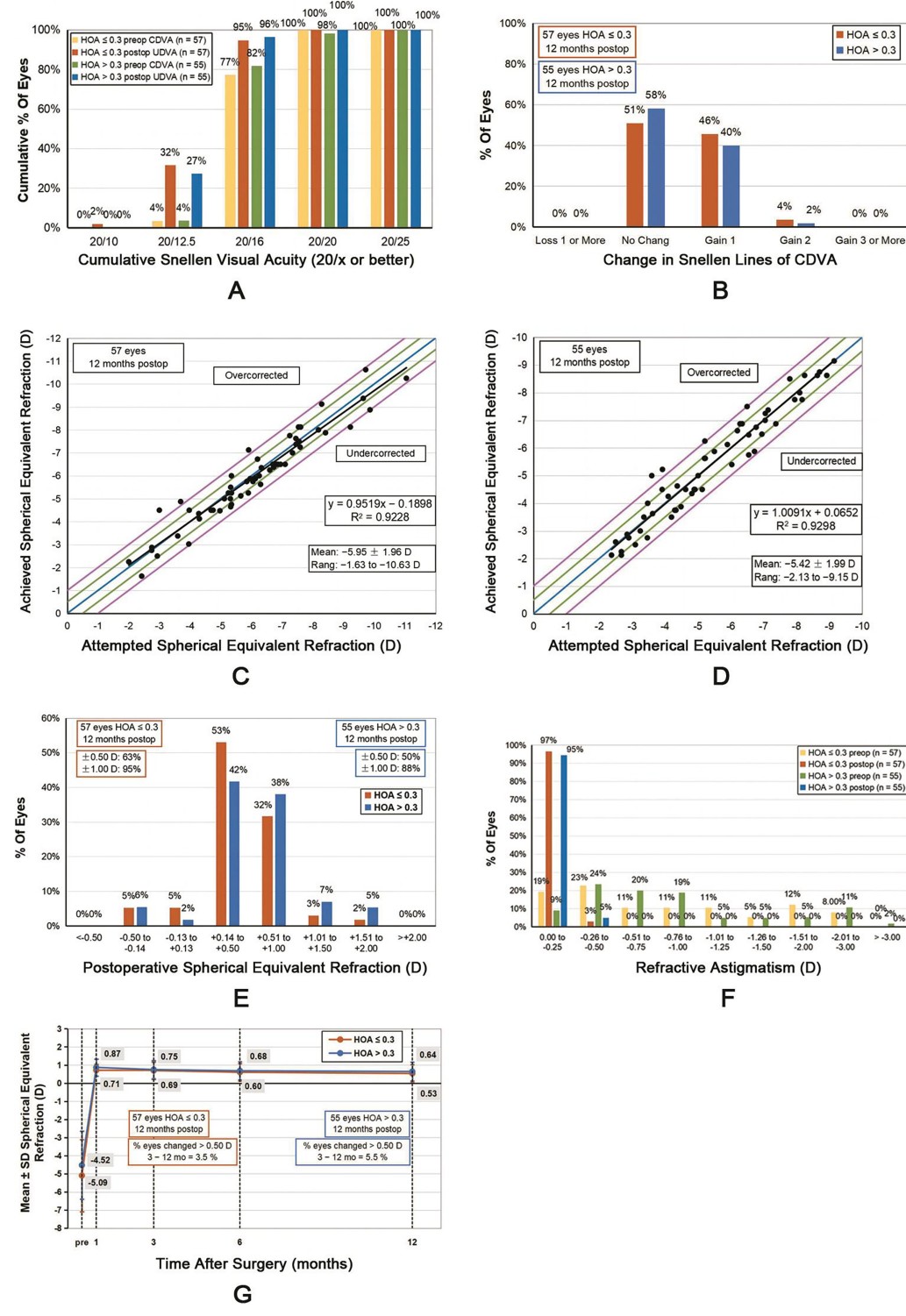

**Figure 1 Standard graphs comparing the HOAs ≤ 0.3 and HOAs > 0.3 groups.** (A) Cumulative 12-month postoperative uncorrected distance visual acuity (UDVA) and preoperative corrected distance visual acuity (CDVA) of both groups. (B) Changes in the CDVA of the two groups at 12 months. (C) Distribution of the achieved spherical equivalent refraction compared with the attempted spherical equivalent refraction of the HOAs ≤ 0.3 group at 12 months. (D) Distribution of the achieved spherical equivalent refraction compared with attempted spherical equivalent refraction of the HOAs > 0.3 group

**Figure 1** (continued)
at 12 months. (E) Comparison of the 12-month postoperative spherical equivalent refractive accuracy of both groups. (F) Comparison of the 12-month postoperative and preoperative refractive astigmatism of both groups. (G) Changes in the spherical equivalent refraction over time.

**Table 2 Comparison of corneal aberrations between the HOAs ≤ 0.3 group and the HOAs > 0.3 group.**

|  | HOA ≤ 0.3 group | HOA > 0.3 group | P value |
|---|---|---|---|
| Number of eyes | 57 | 55 | – |
| SR |  |  |  |
| Preoperative | 0.12 ± 0.07 | 0.12 ± 0.05 | 0.653 |
| Postoperative month 1 | 0.16 ± 0.04* | 0.15 ± 0.05* | 0.169 |
| Postoperative month 3 | 0.16 ± 0.04* | 0.15 ± 0.04* | 0.377 |
| Postoperative month 6 | 0.17 ± 0.05* | 0.16 ± 0.04* | 0.303 |
| Postoperative month 12 | 0.16 ± 0.04* | 0.15 ± 0.04* | 0.285 |
| ΔSR (1 mo–Preop) | 0.04 ± 0.07 | 0.03 ± 0.07 | 0.634 |
| Total corneal aberrations (μm) |  |  |  |
| Preoperative | 1.23 ± 0.61 | 1.29 ± 0.75 | 0.636 |
| Postoperative month 1 | 1.13 ± 0.38 | 1.12 ± 0.28 | 0.802 |
| Postoperative month 3 | 1.12 ± 0.34 | 1.10 ± 0.29 | 0.732 |
| Postoperative month 6 | 1.13 ± 0.36 | 1.11 ± 0.27 | 0.813 |
| Postoperative month 12 | 1.10 ± 0.34 | 1.08 ± 0.24 | 0.715 |
| ΔTCAs (1 mo–Preop) | −0.10 ± 0.54 | −0.18 ± 0.75 | 0.534 |
| Astigmatism (μm) |  |  |  |
| Preoperative | 1.14 ± 0.66 | 1.18 ± 0.79 | 0.761 |
| Postoperative month 1 | 0.63 ± 0.32* | 0.61 ± 0.35* | 0.788 |
| Postoperative month 3 | 0.60 ± 0.28* | 0.62 ± 0.34* | 0.788 |
| Postoperative month 6 | 0.57 ± 0.28* | 0.62 ± 0.29* | 0.426 |
| Postoperative month 12 | 0.57 ± 0.24* | 0.56 ± 0.31* | 0.868 |
| ΔAstigmatism (1 mo–Preop) | −0.51 ± 0.58 | −0.57 ± 0.73 | 0.637 |
| HOAs (μm) |  |  |  |
| Preoperative | 0.38 ± 0.09 | 0.44 ± 0.12 | 0.005** |
| Postoperative month 1 | 0.91 ± 0.32* | 0.86 ± 0.26* | 0.414 |
| Postoperative month 3 | 0.90 ± 0.32* | 0.86 ± 0.26* | 0.415 |
| Postoperative month 6 | 0.94 ± 0.33* | 0.88 ± 0.26* | 0.303 |
| Postoperative month 12 | 0.91 ± 0.32* | 0.87 ± 0.24* | 0.393 |
| ΔHOAs (1 mo–Preop) | 0.53 ± 0.33 | 0.42 ± 0.30 | 0.090 |
| Coma (μm) |  |  |  |
| Preoperative | 0.22 ± 0.21 | 0.24 ± 0.14 | 0.579 |
| Postoperative month 1 | 0.41 ± 0.22* | 0.43 ± 0.25* | 0.673 |
| Postoperative month 3 | 0.43 ± 0.23* | 0.42 ± 0.24* | 0.815 |
| Postoperative month 6 | 0.46 ± 0.25* | 0.46 ± 0.23* | 0.943 |

| | HOA ≤ 0.3 group | HOA > 0.3 group | *P* value |
|---|---|---|---|
| Postoperative month 12 | 0.44 ± 0.26* | 0.43 ± 0.20* | 0.782 |
| ΔComa (1 mo–Preop) | 0.19 ± 0.31 | 0.19 ± 0.28 | 0.991 |
| Trefoil (μm) | | | |
| Preoperative | 0.15 ± 0.07 | 0.18 ± 0.10 | 0.029** |
| Postoperative month 1 | 0.22 ± 0.14* | 0.21 ± 0.11 | 0.721 |
| Postoperative month 3 | 0.21 ± 0.14* | 0.21 ± 0.13 | 0.986 |
| Postoperative month 6 | 0.21 ± 0.15* | 0.19 ± 0.08 | 0.383 |
| Postoperative month 12 | 0.20 ± 0.12* | 0.19 ± 0.10 | 0.669 |
| ΔTrefoil (1 mo–Preop) | 0.08 ± 0.14 | 0.03 ± 0.13 | 0.076 |
| Spherical aberration (μm) | | | |
| Preoperative | 0.21 ± 0.07 | 0.21 ± 0.09 | 0.769 |
| Postoperative month 1 | 0.67 ± 0.27* | 0.59 ± 0.21* | 0.099 |
| Postoperative month 3 | 0.66 ± 0.25* | 0.60 ± 0.21* | 0.131 |
| Postoperative month 6 | 0.68 ± 0.26* | 0.62 ± 0.21* | 0.187 |
| Postoperative month 12 | 0.66 ± 0.25* | 0.63 ± 0.22* | 0.396 |
| ΔSA (1 mo–Preop) | 0.45 ± 0.25 | 0.38 ± 0.21 | 0.103 |

**Note:**
HOAs, higher-order aberrations; SR, strehl ratio; TCAs, Total corneal aberrations; SA, Spherical aberration; Δ, change. Mean ± standard deviation. *Significantly different between preoperative and postoperative values. **Significantly different between the two groups. *P* < 0.05 statistically significant.

**Table 3 Comparison of ocular aberrations between the HOAs ≤ 0.3 group and the HOAs > 0.3 group.**

| | HOA ≤ 0.3 group | HOA > 0.3 group | *P* Value |
|---|---|---|---|
| Number of eyes | 57 | 55 | – |
| Total ocular aberrations (μm) | | | |
| Preoperative | 7.25 ± 2.20 | 6.76 ± 2.30 | 0.253 |
| Postoperative month 1 | 1.28 ± 0.66* | 1.18 ± 0.48* | 0.349 |
| Postoperative month 3 | 1.31 ± 0.64* | 1.21 ± 0.44* | 0.345 |
| Postoperative month 6 | 1.43 ± 0.60* | 1.29 ± 0.45* | 0.152 |
| Postoperative month 12 | 1.50 ± 0.63* | 1.34 ± 0.58* | 0.172 |
| ΔTOAs (1 mo–Preop) | −5.97 ± 2.12 | −5.58 ± 2.32 | 0.359 |
| Defocus (μm) | | | |
| Preoperative | 7.05 ± 2.36 | 6.65 ± 2.32 | 0.367 |
| Postoperative month 1 | 0.88 ± 0.79* | 0.76 ± 0.63* | 0.375 |
| Postoperative month 3 | 0.91 ± 0.76* | 0.78 ± 0.61* | 0.311 |
| Postoperative month 6 | 1.10 ± 0.72* | 0.91 ± 0.61* | 0.125 |
| Postoperative month 12 | 1.15 ± 0.77* | 0.97 ± 0.73* | 0.217 |
| Δ Defocus (1 mo–Preop) | −6.17 ± 2.33 | −5.89 ± 2.40 | 0.533 |
| Astigmatism (μm) | | | |
| Preoperative | 0.87 ± 0.60 | 0.94 ± 0.74 | 0.570 |
| Postoperative month 1 | 0.47 ± 0.25* | 0.38 ± 0.20* | 0.043** |

(Continued)

| Table 3 (continued) | | | |
|---|---|---|---|
| | HOA ≤ 0.3 group | HOA > 0.3 group | *P* Value |
| Postoperative month 3 | 0.43 ± 0.25* | 0.39 ± 0.21* | 0.340 |
| Postoperative month 6 | 0.43 ± 0.21* | 0.37 ± 0.21* | 0.165 |
| Postoperative month 12 | 0.45 ± 0.24* | 0.38 ± 0.21* | 0.112 |
| ΔAstigmatism (1 mo–Preop) | −0.40 ± 0.56 | −0.56 ± 0.69 | 0.177 |
| HOAs (μm) | | | |
| Preoperative | 0.23 ± 0.50 | 0.38 ± 0.06 | <0.001** |
| Postoperative month 1 | 0.61 ± 0.19* | 0.67 ± 0.17* | 0.138 |
| Postoperative month 3 | 0.64 ± 0.20* | 0.67 ± 0.17* | 0.413 |
| Postoperative month 6 | 0.64 ± 0.21* | 0.67 ± 0.19* | 0.446 |
| Postoperative month 12 | 0.66 ± 0.20* | 0.67 ± 0.17* | 0.802 |
| ΔHOAs (1 mo–Preop) | 0.39 ± 0.19 | 0.29 ± 0.18 | 0.006** |
| Coma (μm) | | | |
| Preoperative | 0.12 ± 0.06 | 0.24 ± 0.09 | <0.001** |
| Postoperative month 1 | 0.41 ± 0.19* | 0.44 ± 0.20* | 0.520 |
| Postoperative month 3 | 0.44 ± 0.20* | 0.45 ± 0.18* | 0.686 |
| Postoperative month 6 | 0.42 ± 0.22* | 0.45 ± 0.21* | 0.465 |
| Postoperative month 12 | 0.44 ± 0.22* | 0.45 ± 0.20* | 0.774 |
| ΔComa (1 mo–Preop) | 0.30 ± 0.19 | 0.20 ± 0.21 | 0.012** |
| Trefoil (μm) | | | |
| Preoperative | 0.10 ± 0.06 | 0.16 ± 0.06 | < 0.001** |
| Postoperative month 1 | 0.13 ± 0.08* | 0.16 ± 0.09 | 0.046** |
| Postoperative month 3 | 0.15 ± 0.09* | 0.16 ± 0.10 | 0.434 |
| Postoperative month 6 | 0.15 ± 0.09* | 0.15 ± 0.09 | 0.626 |
| Postoperative month 12 | 0.14 ± 0.08* | 0.15 ± 0.09 | 0.585 |
| Δ Trefoil (1 mo–Preop) | 0.03 ± 0.09 | 0.00 ± 0.11 | 0.149 |
| Spherical aberration (μm) | | | |
| Preoperative | 0.09 ± 0.10 | 0.16 ± 0.08 | <0.001** |
| Postoperative month 1 | 0.32 ± 0.17* | 0.34 ± 0.17* | 0.528 |
| Postoperative month 3 | 0.34 ± 0.17* | 0.35 ± 0.17* | 0.572 |
| Postoperative month 6 | 0.34 ± 0.18* | 0.35 ± 0.18* | 0.835 |
| Postoperative month 12 | 0.36 ± 0.18* | 0.36 ± 0.17* | 0.895 |
| ΔSA (1 mo–Preop) | 0.23 ± 0.20 | 0.18 ± 0.16 | 0.185 |

**Note:**
HOAs, higher-order aberrations; TOAs, total ocular aberrations; SA, spherical aberration; Δ, change. Mean ± standard deviation. *Significantly different between preoperative and postoperative values. **Significantly different between the two groups. *P* < 0.05 statistically significant.

differences were observed in the RMS of TOAs, defocus, astigmatism, HOAs, coma, trefoil, and spherical aberrations between both groups at 3, 6, and 12 months, postoperatively.

The induced ocular HOAs and coma were significantly lower in the HOAs > 0.3 group than in the HOAs ≤ 0.3 group (ΔHOAs: 0.39 ± 0.19 *vs.* 0.29 ± 0.18 μm, *t* = 2.797, *P* = 0.006; Δcoma: 0.30 ± 0.19 *vs.* 0.20 ± 0.21 μm, *t* = 2.542, *P* = 0.012). Pearson's correlation analysis revealed that ΔHOAs were positively correlated with the preoperative RMS of TOAs and

**Table 4 Correlation analysis of ΔHOAs in both groups.**

| | ΔHOAs in the HOAs ≤ 0.3 group | | ΔHOAs in the HOAs > 0.3 group | |
| --- | --- | --- | --- | --- |
| | r-value | P-value | r-value | P-value |
| Preoperative TOAs | 0.527 | <0.001* | 0.314 | 0.020* |
| Preoperative HOAs | −0.045 | 0.742 | −0.315 | 0.019* |
| Preoperative \|sphere\| | 0.515 | <0.001* | 0.305 | 0.023* |
| Preoperative \|cylinder\| | 0.244 | 0.067 | −0.077 | 0.579 |
| Preoperative \|SE\| | 0.506 | <0.001* | 0.274 | 0.043* |

Note:
HOAs, higher-order aberrations; TOAs, total ocular aberrations; Δ, change (1 mo − Preop); |sphere|, the absolute value of sphere; |cylinder|, the absolute value of cylinder; |SE|, the absolute value of spherical equivalent. *A P-value of <0.05 was considered statistically significant.

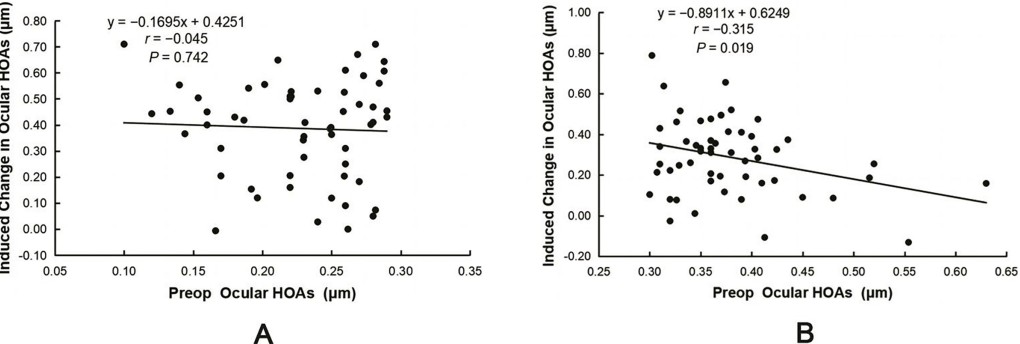

**Figure 2 Correlation between induced changes in ocular HOAs (ΔHOAs) and preoperative ocular HOAs.** (A) No correlation was observed between ΔHOAs and the preoperative RMS of ocular HOAs in the HOAs ≤ 0.3 group. (B) ΔHOAs were negatively correlated with the preoperative RMS of ocular HOAs in the HOAs > 0.3 group.

the absolute values of the sphere (|Sphere|) and SE (|SE|) in both groups (all $P < 0.05$, Table 4). However, ΔHOAs were negatively correlated with the preoperative RMS of ocular HOAs in the HOAs > 0.3 group ($r = -0.315$, $P = 0.019$, Fig. 2B). Lastly, no correlation was observed between ΔHOAs and the preoperative RMS of ocular HOAs in the HOAs ≤ 0.3 group ($r = -0.045$, $P = 0.742$, Fig. 2A).

Multiple stepwise regression analysis was performed to determine the factors influencing ocular ΔHOAs. The explanatory variables included preoperative |Sphere|, |cylinder|, and |SE| and the RMS of preoperative TOAs and ocular HOAs. In the HOAs ≤ 0.3 group, only |SE| ($b = 0.053$, $\beta = 0.532$, $P < 0.001$) significantly and positively predicted ocular ΔHOAs. The change in ΔHOAs, a dependent variable, of 28.3% can be explained by |SE| ($R^2 = 0.283$), with the following regression equation for Δ HOAs = 0.098 + 0.053|SE| ($F = 21.756$, $P < 0.001$). In the HOAs > 0.3 group, |Sphere| ($b = 0.038$, $\beta = 0.374$, $P = 0.004$) significantly and positively predicted ocular ΔHOAs. However, the RMS of preoperative ocular HOAs ($b = -1.081$, $\beta = -0.382$, $P = 0.003$) significantly and negatively predicted ocular ΔHOAs; 23.4% of the ΔHOAs was explained by these two variables ($R^2 = 0.234$).

**Table 5 Comparison of log contrast sensitivity values under photopic, scotopic, and scotopic with glare conditions between the HOAs ≤ 0.3 group and the HOAs > 0.3 group.**

|  | HOA ≤ 0.3 group | HOA > 0.3 group | P Value |
|---|---|---|---|
| Number of eyes | 57 | 55 | – |
| Photopic |  |  |  |
| Preoperative | 1.26 ± 0.11 | 1.24 ± 0.14 | 0.305 |
| Postoperative month 1 | 1.34 ± 0.10* | 1.34 ± 0.10* | 0.939 |
| Postoperative month 3 | 1.37 ± 0.09* | 1.38 ± 0.09* | 0.509 |
| Postoperative month 6 | 1.36 ± 0.10* | 1.40 ± 0.11* | 0.059 |
| Postoperative month 12 | 1.38 ± 0.09* | 1.38 ± 0.10* | 0.883 |
| Scotopic |  |  |  |
| Preoperative | 1.25 ± 0.10 | 1.22 ± 0.12 | 0.127 |
| Postoperative month 1 | 1.30 ± 0.10 | 1.32 ± 0.10* | 0.498 |
| Postoperative month 3 | 1.31 ± 0.09* | 1.34 ± 0.09* | 0.060 |
| Postoperative month 6 | 1.34 ± 0.09* | 1.33 ± 0.10* | 0.807 |
| Postoperative month 12 | 1.33 ± 0.09* | 1.33 ± 0.11* | 0.946 |
| Scotopic with glare |  |  |  |
| Preoperative | 1.17 ± 0.15 | 1.13 ± 0.18 | 0.251 |
| Postoperative month 1 | 1.26 ± 0.11* | 1.26 ± 0.16* | 0.938 |
| Postoperative month 3 | 1.30 ± 0.11* | 1.31 ± 0.09* | 0.442 |
| Postoperative month 6 | 1.31 ± 0.09* | 1.33 ± 0.10* | 0.227 |
| Postoperative month 12 | 1.33 ± 0.09* | 1.31 ± 0.09* | 0.348 |

Note:
HOAs, higher-order aberrations; AULCSF, area under the log contrast sensitivity function. *Significantly different between preoperative and postoperative contrast sensitivity values. Mean ± standard deviation. $P < 0.05$ statistically significant.

The regression equation for ΔHOAs = 0.534 − 1.081HOAs + 0.038|Sphere| ($F = 7.954$, $P = 0.001$).

## CS

AULCSF reflects the overall changes in CS. Table 5 present the AULCSF values under photopic, scotopic, and scotopic with glare conditions. In both groups, after WFG FS-LASIK, the AULCSF values were significantly higher than those preoperatively (all $P < 0.05$). The postoperative AULCSF values gradually improved from 1 month to 6 months in both groups (Fig. 3). No significant differences were observed in all AULCSF values between both groups ($P > 0.05$).

## DISCUSSION

In this retrospective study, we elucidated visual acuity, manifest refraction, corneal SR, the RMS values of corneal and ocular aberrations, and contrast sensitivity function to compare the visual quality after wavefront-guided femtosecond LASIK (WFG FS-LASIK) in patients with different levels of preoperative total ocular higher-order aberrations. We observed that both groups exhibited equivalent and excellent UDVA and manifest refraction at 12 months after WFG FS-LASIK ($P > 0.05$). WFG FS-LASIK procedure

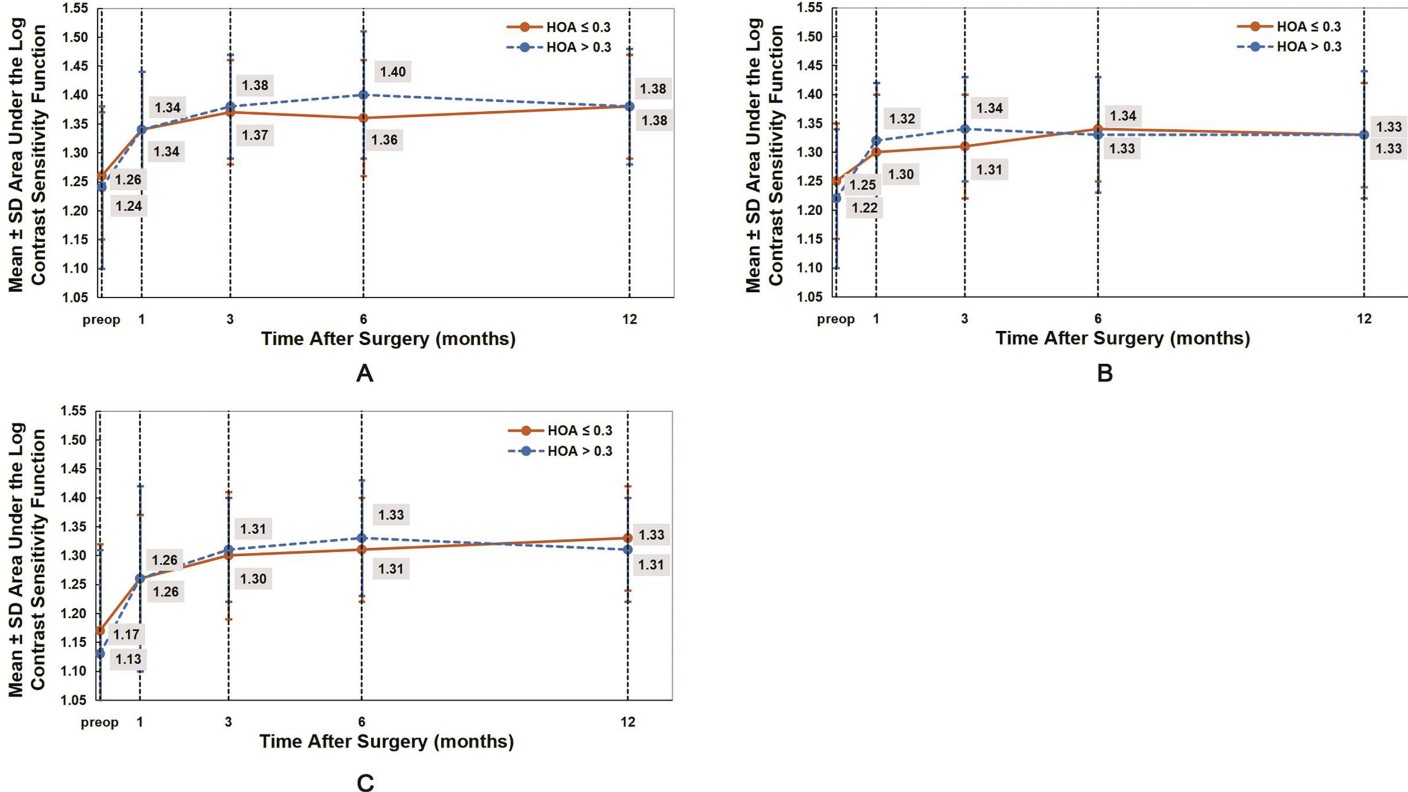

**Figure 3 Comparison of the area under the log contrast sensitivity function (AULCSF) under photopic (A), scotopic (B), and scotopic with glare (C) conditions between the HOAs ≤ 0.3 and HOAs > 0.3 groups.** The postoperative AULCSF values in both groups were significantly higher than those before WFG FS-LASIK. No significant difference was observed between both groups ($P > 0.05$).

showed good efficacy, safety, stability, and predictability for the correction of myopia and myopic astigmatism which was similar to previous studies (*Roe & Manche, 2019*; *Chiang, Valerio & Manche, 2022*). In Fig. 1, the decline in refractive power is more pronounced in the HOAs ≤ 0.3 group although there was no statistically significant difference between the two groups. Further observation and research are needed to determine the decreasing trend of refractive index in the HOAs ≤ 0.3 group. Regular postoperative follow-up and timely adjustment of treatment plan.

Several studies have reported an increase in corneal and ocular aberrations after corneal refractive surgery (*Zhao et al., 2021*; *Zhang et al., 2020*; *Du et al., 2021*; *Sia et al., 2021*; *Gulmez, Tekce & Kamıs, 2020*). *Zhao et al. (2021)* reported that the preoperative RMS of ocular HOAs was 0.422 ± 0.216 μm, which increased to 0.693 ± 0.387 μm after WFG LASIK. In the present study, ocular ΔHOAs were similar to those reported in previous studies (0.39 ± 0.19 and 0.29 ± 0.18 μm for the HOAs ≤ 0.3 and HOAs > 0.3 groups). Furthermore, Pearson's correlation analysis revealed that ΔHOAs are positively correlated with the preoperative RMS of TOAs, |Sphere|, and |SE| (Table 4). *Gui et al. (2021)* reported that the induced HOAs of the WFG combined with high myopia and WFG combined with low and moderate myopia groups were 0.227 ± 0.123 and 0.103 ± 0.203 μm, respectively. This finding indicates that a high degree of myopia can result in more induced HOAs than

low and moderate degrees of myopia. In the present study, although the difference in preoperative SE was not statistically significant between both groups, it exhibited a slightly increased trend in the HOAs ≤ 0.3 group (−5.09 ± 1.99 and −4.52 ± 1.8 D for the HOAs ≤ 0.3 and HOAs > 0.3 groups, $P = 0.121$).This may be one of the reasons why the HOAs ≤ 0.3 group has more induced HOAs than the HOAs > 0.3 group.

We also observed that preoperative HOAs affected postoperative ocular ΔHOAs. The values of ocular ΔHOAs and Δcoma were significantly lower in the HOAs > 0.3 group than in the HOAs ≤ 0.3 group. Pearson's correlation and multiple stepwise regression analyses revealed that ocular ΔHOAs are negatively correlated with the preoperative values of ocular HOAs in the HOAs > 0.3 group ($r = −0.315$, $P = 0.019$). However, this correlation was not observed in the HOAs < 0.3 group (Table 4). Previous studies have reported that the higher the preoperative ocular HOAs, the lower the ΔHOAs. *Jahadi Hosseini, Abtahi & Khalili (2016)* reported that when preoperative ocular HOAs were <0.29 μm, the mean change in ocular HOAs was 0.23 ± 0.18 μm in the LASIK group; however, when the baseline ocular HOAs were ≥0.29 μm, the mean change in value was 0.06 ± 0.13 μm. Furthermore, *Padmanabhan et al. (2008)* reported that induced changes in ocular HOAs were weakly and negatively correlated with their preoperative values ($r = −0.37$, $P = 0.06$). Therefore, some researchers believe most patients with a preoperative RMS HOA of <0.30 μm do not need to undergo WFG LASIK and that wavefront-guided technology provides the greatest benefit for patients with larger preoperative HOA values (*Stonecipher & Kezirian, 2008*; *Stonecipher, Parrish & Stonecipher, 2018*; *Zhang et al., 2013*).

Although the ocular ΔHOA and Δcoma values were lower in the HOAs > 0.3 group than in the HOAs ≤ 0.3 group, the difference in ocular and corneal aberrations was not statistically significant between both groups during follow-up after WFG FS-LASIK (Tables 2 and 3). This indicates that postoperative corneal and ocular aberrations are consistent, regardless of whether the proportion of preoperative ocular HOAs is high or low. In contrast, *Zhang et al. (2013)* have reported a significant negative correlation between induced changes in ocular HOAs and their preoperative values in both the WFG LASIK and conventional LASIK groups ($r = −0.577$, $P < 0.001$; $r = −0.443$, $P < 0.001$, respectively). This finding indicates that ocular ΔHOAs are affected by preoperative HOAs either after WFG LASIK or conventional LASIK. As the preoperative factors such as sphere, SE, and the RMS of TOAs and ocular HOAs can influence the changes in HOAs after WFG FS-LASIK. Prior to surgery, it is necessary to evaluate the ocular higher-order aberrations in order to better predict surgical outcomes. Choose different surgical plans based on different types of eyes to minimize changes in higher-order aberrations.

CSF is a crucial index to evaluate visual quality after corneal refractive surgery (*Ryan et al., 2018*; *He et al., 2022*; *Shao et al., 2022*). Several studies have reported that CSF values slightly decrease in the early stage after conventional LASIK and then gradually recover to the preoperative level; however, they have also reported that CSF values after WFG FS-LASIK are higher than those after conventional LASIK, and the postoperative CSF values are significantly higher than their preoperative values in the early stage after WFG FS-LASIK (*Zhang et al., 2013*, *2008*). In the present study, the AULCSF values of the two groups were significantly improved at 1 month postoperatively (Fig. 3). No significant
differences were observed in the postoperative CSF values at all AULCSF values between both groups (Table 5); this indicates that the visual quality after WFG FS-LASIK was significantly improved in both groups, regardless of whether the RMS of preoperative total ocular HOAs is high or low.

The reasons for improved CSF after WFG FS-LASIK may be as follows. (1) Postoperative UDVA improved, even better than its preoperative CDVA. In the present study, the gradual increase in AULCSF values in the early postoperative period was consistent with the improvements in UDVA in the corresponding period. (2) Improvement in corneal SR: Point spread function (PSF) is an important index for objectively evaluating image quality and is related to diffraction, aberration, and scattering (Charman, 2005). SR, a quantitative optical index of PSF, is widely used to evaluate visual quality. A higher SR value indicates better visual quality (Chandra et al., 2022; Chen et al., 2023; Liu et al., 2019). Tuan, Chernyak & Feldman (2006) has reported that patients with night vision complaints after LASIK have significantly lower SR values than those without any complaints after LASIK. In the present study, the SR values of both groups were significantly increased after WFG FS-LASIK; this change will be beneficial in creating a clearer image. (3) Postoperative aberration significantly decreased: Compared with the preoperative results, postoperative lower-order aberrations significantly decreased. However, the TCAs did not significantly decrease because the decrease in corneal lower-order aberrations (astigmatism) was offset by an increase in HOAs (Table 2). Therefore, we hypothesize that visual quality is less affected by changes in TCAs. Our results suggest that although postoperative HOAs increased, with a significant decrease in lower-order aberrations, TOAs decreased to approximately 1/6 of the preoperative values. Therefore, the significant decrease in TOAs significantly contributes to improving postoperative CS.

The limitation of the present study are that it was a retrospective study. Due to the limited number of participants included in the study, we refrained from conducting a grouping analysis based on different levels of myopia. Nonetheless, the current study is valuable in that we evaluated the visual quality after WFG FS-LASIK in patients with different levels of preoperative total ocular higher-order aberrations. Due to the excellent visual quality after WFG-LASIK, WFG FS-LASIK may provide significant visual benefits for a wider range of patients, even those with relatively low preoperative ocular HOAs. Future prospective studies with larger samples and longer follow-up periods are needed to further validate these findings and explore the long-term visual outcomes and the potential impact of other factors on visual quality after WFG FS-LASIK. Whether the visual quality after WFG FS-LASIK and other refractive surgery is different due to different levels of preoperative ocular HOAs also warrants further investigation.

## CONCLUSION

In conclusion, the ocular $\Delta$HOAs and $\Delta$coma were lower in the HOAs > 0.3 group than in the HOAs ≤ 0.3 group. However, both groups achieved equivalent and excellent visual quality after WFG FS-LASIK. WFG FS-LASIK may provide significant visual benefits for a wider range of patients.

### Funding

This work was funded by the Key Program of Chongqing Natural Science Foundation. China (No. cstc2021ycjh-bgzxm0064). The funders had no role in study design, data collection and analysis, decision to publish, or preparation of the manuscript.

### Grant Disclosures

The following grant information was disclosed by the authors:
Chongqing Natural Science Foundation China: cstc2021ycjh-bgzxm0064.

### Competing Interests

The authors declare that they have no competing interests.

### Author Contributions

- Yu Zhang conceived and designed the experiments, analyzed the data, prepared figures and/or tables, authored or reviewed drafts of the article, and approved the final draft.
- Yangrui Du performed the experiments, prepared figures and/or tables, authored or reviewed drafts of the article, and approved the final draft.
- Ming He performed the experiments, authored or reviewed drafts of the article, and approved the final draft.
- Youdan Zhang analyzed the data, authored or reviewed drafts of the article, and approved the final draft.
- Zhiyu Du conceived and designed the experiments, authored or reviewed drafts of the article, and approved the final draft.

### Human Ethics

The following information was supplied relating to ethical approvals (*i.e.*, approving body and any reference numbers):

This study was approved by the Ethics Committee of the Second Affiliated Hospital of Chongqing Medical University (No. 76/2022). Patients were informed about study inclusion and provided written informed consent.

### Data Availability

The raw measurements are available in the Supplemental Files.

### Supplemental Information

Supplemental information for this article can be found online at http://dx.doi.org/10.7717/peerj.17940#supplemental-information.

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
