# Peer review of "Comparison of visual quality after wavefront-guided LASIK in patients with different levels of preoperative total ocular higher-order aberrations: a retrospective study"

_PeerJ, doi:10.7717/peerj.17940_

## Round 0.1 · original submission · Major Revisions

After careful consideration and review by three experts in the field, we have decided to invite you to revise and resubmit your manuscript. Please address all the reviewers' comments thoroughly and provide a point-by-point response with your resubmission. Pay particular attention to the major concerns raised.

Reviewer 1 ·

Basic reporting

This retrospective clinical research article compared visual quality after wavefront-guided LASIK in myopic patients with different levels of total ocular higher-order aberrations. The English written and text were acceptable.Literature references included sufficient introduction and background.

Experimental design

This original primary research is within aims and scope of the journal.

Validity of the findings

Almost all the data have been provided and support the conclusions.

Additional comments

There are some flaws:
1. "high-order aberrations" should be replaced by "higher-order aberrations".
2. Since the object of this research aimed for visual quality, the patient's subjective visual scale should also be mentioned.
3. In Figure 1, from graph C we can find the undercorrection tendency. But the graph E showed overcorrection.
4. The amount of pre-operational spherical equivalent presented in Figure 1 and table 1 is inconsistent.

Reviewer 2 ·

Basic reporting

1. In line 54-55 and line 192, there is light gray highlight when look carefully. And I am not sure if there are other lines. Please check it.
2. Introduction section seems insufficient. You told the advantages of WFG FS-LASIK over conventional LASIK. Maybe you can add some other contents, like what is its principle for minimizing the induction of HOA?
3. You said that “even some studies have suggested that the wavefront-guided technology is not needed for patients with preoperative ocular HOAs of <0.30 μm”, then “therefore, what about visual quality after WFG FS-LASIK for patients with preoperative ocular HOAs of<0.30 μm? ” So what’s the reason for their suggestion? Is it the poor visual quality? There is no causal relationship between these two sentences.

Experimental design

1. You explained detailedly why you choose <0.30 μm as the grouping standard, and it’s good. But it is suggested in the Patients and method section rather than in the Results section. Please think about it.
2. I don’t think this is a case-control study. It’s more like a retrospective cohort study. Please confirm it.

Validity of the findings

1. It is not recommended to repeat the results in the Discussion section, for example, in the first paragraph of Discussion. Please simplify it.
2. Conclusions are stated, but they should have been better and should be tightly linked to your original research question promoted in the Introduction. For example, how can the findings of this study help us choose appropriate refractive surgery to correct high aberrations? And whether the wavefront-guided technology is needed or not for patients with preoperative ocular HOAs of >0.30 μm?Please think about it.
3.The content in Acknowledgements section is not necessary.

Additional comments

None

Reviewer 3 ·

Basic reporting

This research question aims to determine if the visual quality achieved with wavefront-guided LASIK differs in patients with varying levels of preoperative total ocular HOAs. Overall, this study demonstrates the effectiveness of WFG-LASIK in correcting HOAs, particularly in patients with high preoperative HOAs. It also highlights the importance of considering preoperative HOAs and spherical equivalent when predicting outcomes. Further research is needed to confirm these findings and explore the underlying mechanisms. There are still many areas for improvement in the article.
1. This retrospective study demonstrates that while WFG FS-LASIK is more effective at reducing HOAs in patients with higher preoperative levels of HOAs ( > 0.3 µm), both groups achieved equivalent and excellent visual quality outcomes. This suggests that WFG FS-LASIK may provide significant visual benefits for a wider range of patients, even those with relatively low preoperative HOAs. However, the retrospective design and sample size limit the generalizability of these findings. Future prospective studies with larger samples and longer follow-up periods are needed to further validate these findings and explore the long-term visual outcomes and the potential impact of other factors on visual quality after WFG FS-LASIK.
2. In the Purpose of the Abstract, please highlight the clinical relevance: Instead of just stating the comparison, emphasize why this comparison is important for clinicians and patients. Such as "To compare the visual quality after wavefront-guided femtosecond LASIK (WFG FS-LASIK) in patients with different levels of preoperative total ocular high-order aberrations to guide clinical decision-making regarding patient selection and treatment strategies."
3. Overall, your introduction is well-written and sets the stage for an important research study. The information you've presented provides a strong foundation for exploring the complex relationship between preoperative HOAs and visual quality after WFG FS-LASIK. Some explanations are still need to be added in the Introduction.
1) Consider adding a sentence or two about the specific visual quality metrics you will be evaluating: While you mention "visual quality," be more precise about the specific metrics you will use to assess it (e.g., uncorrected visual acuity, contrast sensitivity, subjective assessments of glare and halos).
2) Briefly mention your study design: While you mention it's a retrospective study, you could also mention if it's a case-control study or a cohort study and briefly explain the data collection methods.
3) Provide a concise statement of your hypothesis: You could conclude the introduction with a clear statement of your hypothesis about the relationship between preoperative HOAs and visual quality after WFG FS-LASIK.
4. In Patients and Methods, Briefly explain why you chose specific procedures and parameters. For example, you could mention why you measured corneal topography, wavefront aberrations, and contrast sensitivity. In addition, please emphasize the relevance of the data you collected to answering your research question.
5. Instead of just stating the reason for the grouping, explicitly explain why it's important for your study. For example: "Given the suggestion that wavefront-guided technology might be more beneficial for patients with higher preoperative HOAs, we divided the patients into two groups based on the mean value of preoperative RMS of ocular HOAs: HOAs ≤ 0.3 µm and HOAs > 0.3 µm. This grouping allowed us to compare the visual outcomes of WFG FS-LASIK in patients with different levels of preoperative aberrations." You could also add a sentence about the rationale for choosing the mean value of 0.3 µm as the cut-off point.
6. In Figure 1, Research has shown that the surgical outcomes of both types of eyeballs perform well in terms of predictability and stability, but the decline in refractive power is more pronounced in the HOAs ≤ 0.3 group. Although the surgical results of the two groups are similar, further observation and research are needed to determine the decreasing trend of refractive index in the HOAs ≤ 0.3 group. Regular postoperative follow-up and timely adjustment of treatment plan.
7. In Figure 2 and 3, the study seems to suggest that WFG FS-LASIK leads to an improvement in ocular aberrations and contrast sensitivity, with some differences observed between groups based on their preoperative HOAs levels. The statistical analyses, including Pearson's correlation and multiple stepwise regression, were used to explore the relationships between preoperative factors and the changes in ocular aberrations postoperatively. The results indicate that preoperative factors such as sphere, cylinder, SE, and the RMS of TOAs and ocular HOAs can influence the changes in HOAs after the procedure. Prior to surgery, it is necessary to evaluate the higher-order aberrations of the eyeball in order to better predict surgical outcomes. Choose different surgical plans based on different types of eyeballs to minimize changes in high-order aberrations.
8. In summary, although this study suggests that WFG FS-LASIK achieves good visual quality for patients with different preoperative high-order aberrations of the eyeball, the limitations of the study also need to be noted. In the future, more large-scale and prospective studies are needed to further validate the safety, efficacy, and long-term effects of WFG FS-LASIK. Such as: Lack of Control Group: The study didn't include a control group of patients who underwent standard LASIK. This makes it difficult to directly compare the outcomes of WFG-LASIK to standard LASIK, which could be useful in assessing the overall benefit of wavefront guidance. Other Visual Quality Factors: While the study focused on HOAs, other factors, such as corneal topography, corneal biomechanics, and ocular surface health, can also significantly impact visual quality after refractive surgery.
9. A large number of descriptions of the results have been added to the discussion, and it is necessary to streamline these descriptions and discuss more about the results.
10. The writing in the paper is generally clear, but there are a few areas where the language could be more concise and academic. The authors should improve their writing and make their research more accessible to a wider audience.

Experimental design

Comments are summarized in Basic reporting.

Validity of the findings

Comments are summarized in Basic reporting.

---

## Round 0.2 · accepted · Accept

Both reviewers have expressed satisfaction with the changes you've made, and they recommend acceptance of your paper. After careful consideration of the reviewers' comments and my own evaluation, I am pleased to inform you that your manuscript has been accepted for publication.

Reviewer 1 ·

Basic reporting

The revised article included sufficient introduction and background to demonstrate the work fits into the broader field of knowledge and demonstrated raw data to support the conclusions.

Experimental design

The research methods described provided sufficient and detail information to replicate.

Validity of the findings

The conclusions are well stated and linked to original research question and supporting results.

Additional comments

The authors of the article entitled “Comparison of visual quality after wavefront-guided LASIK in patients with different levels of preoperative total ocular high-order aberrations” have responded to all the reviewers’ comments and revised accordingly. The constructions and language were improved obviously. I endorse to accept and publish this article.

Reviewer 3 ·

Basic reporting

The authors' revision is careful and meticulous, and has made a good response and revision to the comments, which I think has solved my original concerns.

Experimental design

no comment

Validity of the findings

no comment